# An Accurate and Efficient Segmentation Method of White Matter Hyper-intensity using Deep Learning

**Hyunwoo Oh**[*1]                                                       HW.OH@VUNO.CO
**Dongsoo Lee**[*1]                                                      DSLEE@VUNO.CO
**Kyuhwan Jung**[1]                                             KHWAN.JUNG@VUNO.CO
[1] *VUNO Inc., Seocho-gu, Seoul, South Korea.*

## Abstract

Unlike the rich study of automatic segmentation of white matter hyperintensity (WMH), only a few divide WMH into Deep WMHs (DWMH) and Periventricular WMHs (PVWMH). Distinguishing the two is critical because of their different clinical implications. In this work, we propose a novel WMH segmentation method which segments both DWMH and PVWMH from T2-FLAIR MR images. The proposed method operates in two steps: WMH separating segmentation and subdivision. In the first step, convolutional neural networks (CNNs) are trained to separate PVWMH from DWMH segmentation, and both masks were combined into one whole WMH mask in the first stage. The whole WMH mask is then divided into DWMH and PVWMH via lateral ventricle segmentation and post-processed through a connected components algorithm. This post-processing adheres to the "continuity to ventricle" clinical criterion for dividing PVWMH and DWMH based on the segmented lateral ventricles. The proposed method not only reduces false negatives, achieving a Dice coefficient score of 0.83, but also partitions WMH into PVWMH and DWMH, enabling a more fine-grained diagnosis.

**Keywords:** Semantic Segmentation, Deep Learning, White Matter Hyperintensities, Connected Components Algorithm, Medical Imaging.

## 1. Introduction

Deep learning has already been successfully applied to medical imaging including brain area parcellation (Henschel et al., 2020; Suh et al., 2020), Alzheimer's disease classification (Gupta et al., 2019), and white matter hyperintensity (WMH) segmentation (Rachmadi et al., 2018). This work is concerned with automating segmentation of WMH from T2 fluid-attenuated inversion recovery (FLAIR) magnetic resonance imaging (MRI) sequences. WMH lesions manifest as increased brightness relative to surrounding areas and are associated with various cardiovascular diseases, cognitive functioning, vascular dementia, and mood disorder (Griffanti et al., 2018). In the MICCAI 2017 WMH challenge, many researchers proposed various methods to segment WMH and they obtained remarkable results (Li et al., 2018). The winner of this challenge uses a fully connected network based on U-Net, data augmentation and ensemble techniques. Nonetheless, existing algorithms perform coarse segmentation on WMH as a whole, and there is yet to be a deep learning algorithm that can separately segment periventricular WMHs (PVWMH) and deep WMHs (DWMH).

---

[*] Contributed equally

This distinction is important in the clinical setting because a fine-grained segmentation is critical features in classifying vascular dementia and alzheimer's disease (Smith et al., 2016).

PVWMH are located adjacent to lateral ventricles (LV), whereas DWMH are the area under the cortex (Van den Heuvel et al., 2006). The two WMH have distinctive functional, histopathological, and aetiological features (Kim et al., 2008). PVWMH are mainly related to impaired cognitive function and reduced cerebral blood flow (ten Dam et al., 2007; Bolandzadeh et al., 2012). DWMH are associated with mood disorders, migraine, and ischemic complications (Cees De Groot et al., 2000; Krishnan et al., 2006). Owing to their clinical implications, the distinction between PVWMH and DWMH is clinically important.

Previous studies developed WMH segmentation methods such as the WMH segmentation using extracted white matter (Ithapu et al., 2014), a machine learning based WMH segmentation (Rachmadi et al., 2017), a deep learning based WMH segmentation (Ghafoorian et al., 2017), only DWMH segmentation using machine learning methods (Park et al., 2018). Recently, many studies used U-Net based architectures to segment WMH and they want to reduce false positives effectively (Hong et al., 2020). However, most of them focus on the whole WMH and are not interested in subdividing WMH into PVWMH and DWMH. Even if they are interested, it is only deep learning based WMH segmentation, not using clinical criteria. Only deep learning based segmentation can lead to incorrect results that do not fit the PVWMH and DWMH definitions. Therefore, we add post-processing step to address this problem using clinical criteria.

In this paper, we propose an accurate and efficient subdividing WMH segmentation pipeline based on clinical criteria called a **"continuity to ventricle"** rule which means that the WMH adjacent to the ventricle surface is PWMH, otherwise it is DWMH (Kim et al., 2008) (Fazekas et al., 1987). First, we trained PVWMH and DWMH segmentation models separately with W-Net (Galdran et al., 2020) which is known as more efficient architecture than U-Net series because of fewer the number of parameters. The reason to differentiate between DWMH and PVWMH is that the number of pixels corresponding to DWMH is small. This may lead to the result that DWMH does not learn well compared to PVWMH. Then, we aggregate both masks into one mask and divide WMH into PVWMH and DWMH via post-processing which apply a "continuity to ventricle" rule using segmented LV masks.

We present our main contributions.

1. We proposed a separate segmentation method for PVWMH and DWMH to reduce false negatives.

2. We make possible to distinguish PVWMH and DWMH by the clinical criteria using segmented lateral ventricles.

3. Our model will help diagnose disease in the clinical setting and can be used for visual rating based on the fazekas scale in the future.

## 2. Methodology

Segmentation of PVWMH and DWMH is extremely difficult because of their high class imbalance as a result of the sparsity of lesions. In this section, we describe a novel method consisting of WMH segmentation and subdivision steps to address this issue.

## 2.1. Distinct Segmentation of Periventricular and Deep WMH

Figure 1: A schematic diagram of WMH segmentation step.

As shown in Fig 1, WMH segmentation step consists of two parts: brain extraction, WMH segmentation. Brain extraction is one of the most important step in many neuro-image researches. We built our in-house brain extraction tool using a rigid transformation and a 3D U-Net based deep learning model. After brain extraction, we trained two W-Net (Galdran et al., 2020) based CNN models each for PVWMH and DWMH to address DWMH's sparsity. The most popular CNN architecture in medical image segmentation is U-Net (Ronneberger et al., 2015). However, U-Net series CNN architectures (Zhou et al., 2018; Huang et al., 2020) have some limitations that it has many parameters and cannot use memory and information efficiently. The little W-Net we used for segmentation reduced the number of parameters and increased the representation power of features by using U-Net twice (Galdran et al., 2020). Therefore, we decide to apply little W-Net. After separately segmenting PVWMH and DWMH, we aggregated two masks into one mask to apply a subdivision step.

WMH segmentation is known as a class imbalanced problem. If training as usual, imbalanced class leads to unreliable training that only the majority class results in. Many methods were studied to address this problem such as weighted cross entropy (CE), Dice coefficient (Zou et al., 2004), Focal loss (Lin et al., 2017), tversky loss (Salehi et al., 2017). In this paper, we adapt the weighted CE loss transformed with log function for solving a class imbalanced problem. We employ a logarithmic transformation to smooth the weight. This loss is computed as class proportions for each batch and this weight is applied for positive class term to alleviate a class imbalanced problem. Equation1 is our loss function and $n_1$ : the number of WMH pixels, n : the number of total pixels, $\alpha$ is a hyperparameter to adjust smoothing degree, $w_1$ : a log-smoothed weight for positive term.

$$Our\ loss = w_1 \times (Y \times -log(Y_{pred})) + (1 - Y) \times (-log(1 - Y_{pred}))$$
$$w_1 = log((\alpha \times n)/n_1)\ (0 < \alpha < 1) \tag{1}$$

## 2.2. Fine-grained Subdivision

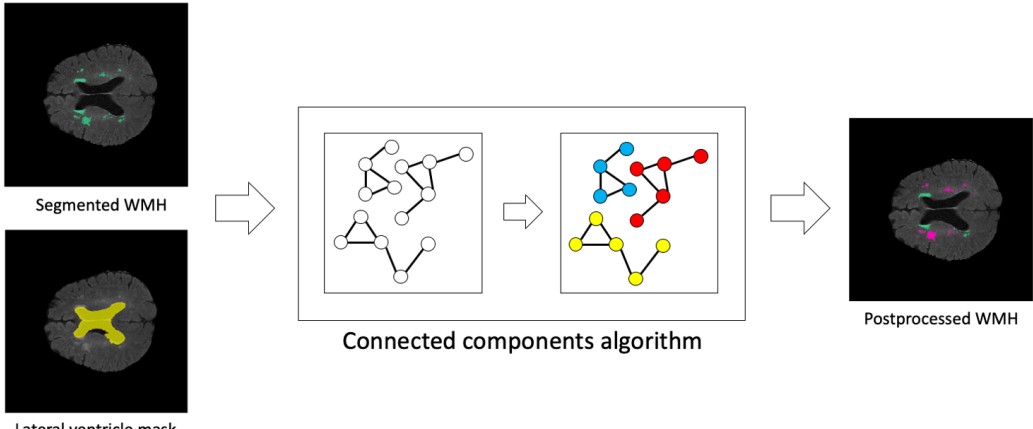

Figure 2: A schematic diagram of WMH subdivision step.

To apply a subdivision step (post-processing step), we need to segment lateral ventricles (LV) masks in T2-FLAIR MRI. Our LV segmentation model was also trained with a W-Net architecture. We dilated segmented LV masks to distinguish PVWMH. To reflect a "continuity to ventricle" rule, we employ a connected components algorithm (Wu et al., 2005) to classify WMH touching the dilated LV mask as PVWMH and the others are DWMH (Fig 2). By applying this step, we can obtain WMH masks segmented by the clinical criteria.

## 3. Experiment

### 3.1. DATASET

60 T2 fluid-attenuated inversion recovery (FLAIR) MRI were manually segmented by an experienced rater using a ITK-SNAP software (Yushkevich et al., 2016). All WMH masks were drawn along the axial space and the masks represent PVWMH and DWMH labels separately. 60 subjects also had T1-weighted MRI scans. 50 subjects were randomly selected as training data and the rest of 10 subjects were used for testing data. All T2-FLAIR and T1-weighted MRI scans have been acquired with a Philips Achieva 3T scanner.

### 3.2. Lateral Ventricle segmentation

#### 3.2.1. PREPROCESSING

We need lateral ventricles (LV) ground truth masks in T2-FLAIR MRI to train a LV segmentation model using T2-FLAIR. To get ground truth masks, brain regions in T1-weighted MRI were segmented by our in-house brain parcellation tool and segmented LV in T1 space from above analysis is co-registered to T2-FLAIR space using a SimpleElastix software (Marstal et al., 2016) (Fig 3). These LV masks are used as the ground truth for training a LV segmentation model using T2-FLAIR as input data. T2-FLAIR preprocessing is the same as the above segmentation model.

### 3.2.2. TRAINING

The LV segmentation model was trained with W-Net which consists of (8,16,32) channels encoder and (16,8,1) channels decoder. The model has been trained for 1000 epochs with a batch size of 16 and learning rate was set to 0.001. We apply a Adam optimizer with hyperparameters fixed to $P_1 = 0.9$, $P_2 = 0.999$ and loss function is logarithmic smoothing weighted binary cross entropy loss which was explained above methodology section. Learning scheduler was CosineAnnealingWarmRestarts which was set to $T_0 = 10$, $T_{mult}$ =1. During training, augmentation techniques including fliplr, flipud, random rotation up to a range of 30 degrees, random x-axis shearing up to a range of 30 degrees were utilized to train the model robustly.

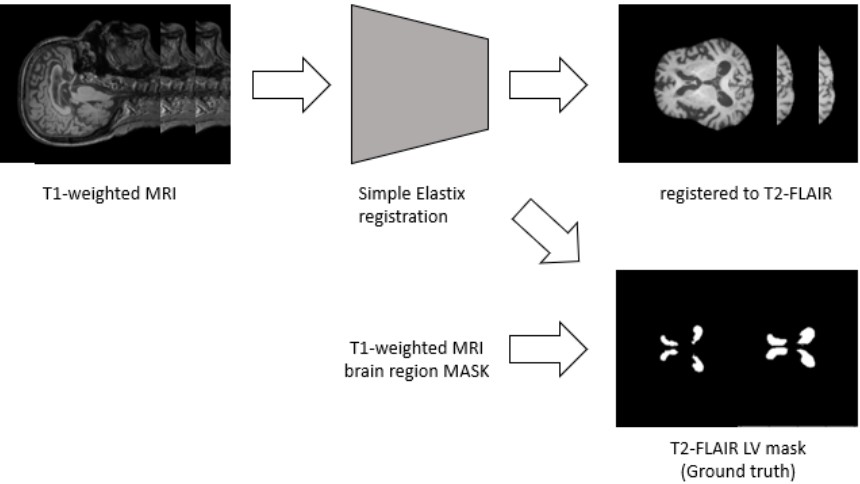

Figure 3: A schematic diagram of LV dataset preprocessing. We apply the same matrix which registeres T1-weighted MRI into T2-FLAIR space to register LV mask.

### 3.3. PV/D WMH segmentation model versus Whole WMH segmentation model

PV/D segmentation model consists of PVWMH segmentation model, DWMH segmentation model, and a process which aggregates separately segmented masks. The PVWMH segmentation model is trained with ground truth which only has PVWMH lesions. Likewise, the DWMH segmentation model is trained with masks which have DWMH lesions. The process of aggregating PVWMH and DWMH is to cover the DWMH mask over the PVWMH mask. Whole WMH segmentation model is trained with whole WMH masks defined as the combining of PVWMH and DWMH as usual.

The W-Net architecture which consists of (8,16,32,64) channels encoder and (32,16,8,1) channels decoder were used for PV/D and whole WMH segmentation. We utilized an Adadelta optimizer with a step learning scheduler by multiplying 0.9 times to a learning rate per 25 epochs and the initial learning rate was set to 0.5. Other hyperparameters and

augmentation techniques are same as the LV segmentation model. Moreover, we trained the whole WMH segmentation models using other architectures to investigate general segmentation performance (Table 1). Our proposed PV/D model shows overall better performance than any other models. Compared with a whole W-Net model, in the PV/D model, the dice coefficient is 2% higher and the precision is slightly lower, but the recall score is significantly improved. This result shows that a PV/D separating method can effectively reduce false negatives. Although U-Net had highest score in precision, it had poor AUPRC compared with the W-Net model. Because of this result, we think that W-Net extracts image features more effectively than U-Net series. Because test dataset is different, it is hard to directly compared the performance of the model with performance of the winner of MICCAI 2017 WMH segmentation challenge (Kuijf et al., 2019). But, our proposed method shows better performance by large margin in dice score than that of the winner of MICCAI 2017 WMH challenge with smaller number of parameters.

Table 1: PV/D and whole WMH segmentation experiment

| Model | Dice | Dice(2d-slice) | Precision | recall | AUPRC | Params |
|-------|------|----------------|-----------|--------|-------|--------|
| W-Net(PV/D) | **0.834** | **0.743 ± 0.161** | 0.815 | **0.853** | - | 0.4M |
| W-Net(whole) | 0.818 | 0.724 ± 0.170 | 0.822 | 0.814 | **0.903** | **0.2M** |
| U-Net(whole) | 0.795 | 0.695 ± 0.222 | **0.868** | 0.734 | 0.819 | 17.2M |
| U-Net++(whole) | 0.742 | 0.582 ± 0.251 | 0.781 | 0.706 | 0.785 | 9.1M |
| MICCAI2017 | 0.80 | - | - | 0.84 | - | 8.7M |

*PV/D: periventricular/deep WMH separately segment models
*whole: whole WMH segmentation model (no separate)
*W-Net(PV/D) model cannot calculate AUPRC becasuse of mask aggregating process
*The model MICCAI2017 is the winner of MICCAI 2017 WMH segmentation challenge

### 3.4. PV/D segmentation with a connected components algorithm

After segmentation, we applied a connected components algorithm to distinguish PVWMH and DWMH based on a "continuity to ventricle" rule. The segmented lateral ventricles were expanded manually using a dilation function in the skimage module (Van der Walt et al., 2014). The degree of expansion can be adjusted according to the subjectivity of the researcher as shown in Figure 4. Then, segmented WMH which reaches expanded LV are classified as PVWMH, and other WMH are classified as DWMH. Figure 5 shows that W-Net output masks have many mis-classification masks when compared with ground truth masks. After applying the post-processing, the mis-classified masks were almost re-classified to its proper class. We think that post-processing step complements DWMH segmentation which is known as a difficult task. It also proves that this step makes possible accurate WMH subgroup segmentation according to the widely used WMH definition and will help to make accurate diagnosis.

## 4. Discussion

The criteria for distinguishing PVWMH and DWMH is somewhat arbitrary and various methods are available. **Continuity to ventricles**: WMH adjacent to the ventricle is

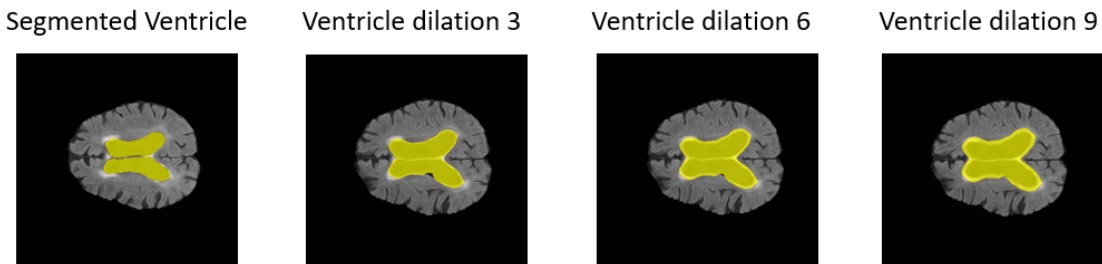

Figure 4: LV mask expansion according to the degree of dilation.

PVWMH, otherwise it is DWMH; **10mm distance**: WMH less than 10mm distance from the ventricles is PWMH, otherwise it is DWMH; **3-13mm distance**: WMH less than 3mm from the ventricles is juxtaventricular, 3-13mm distance from ventricules is PVWMH, otherwise it is DWMH; **Emperical distance**: PVWMH empirically defined, otherwise it is DWMH (Griffanti et al., 2018). Among these, We adapted the "continuity to ventricles" rule which is widely used to distinguish PVWMH and DWMH. Without following clinical rule, there is a problem with PVWMH and DWMH that do not fit the definition. To solve this problem, we apply a WMH subdivision step using the LV segmentation mask. This step can be utilized to all criteria which distinguish PVWMH, DWMH based on the LV segmentation mask, and can reflect the physicians's subjectivity to adjust a LV mask dilation.

Table 1 demonstrates that the PV/D separate segmentation model obtains better performance than the whole segmentation model. We believe that the whole model cannot segment well DWMH because of their sparsity. Since most of WMH are composed of PVWMH, it is difficult to segment sparse DWMH. PV/D models solve DWMH learning problems by allowing the model to focus on DWMH via a separate segmentation. Then, this method increases the recall score by reducing false negatives.

However, this analysis has some limitations. As can be seen from the various criteria defining WMH, it is somewhat arbitrary to distinguish PVWMH and DWMH. Moreover, it needs to test on multi MRI vendors to confirm its consistent segmentation performance.

## 5. Conclusion

We proposed a novel pipeline which consists of a WMH segmentation step and a WMH subdivision step. PV/D separate models improve segmentation performance by effectively reducing false positive pixels. A WMH subdivision step makes it possible to distinguish between PVWMH and DWMH by the clinical criteria. We believe that this method can be useful to divide the subgroup WMH classification accurately and it will help clinical diagnoses such as vascular dementia, alzheimer's disease. In the near future, we anticipate extending this work to classify fazekas scale for each PVWMH, DWMH using multi-task learning (MTL) methods.

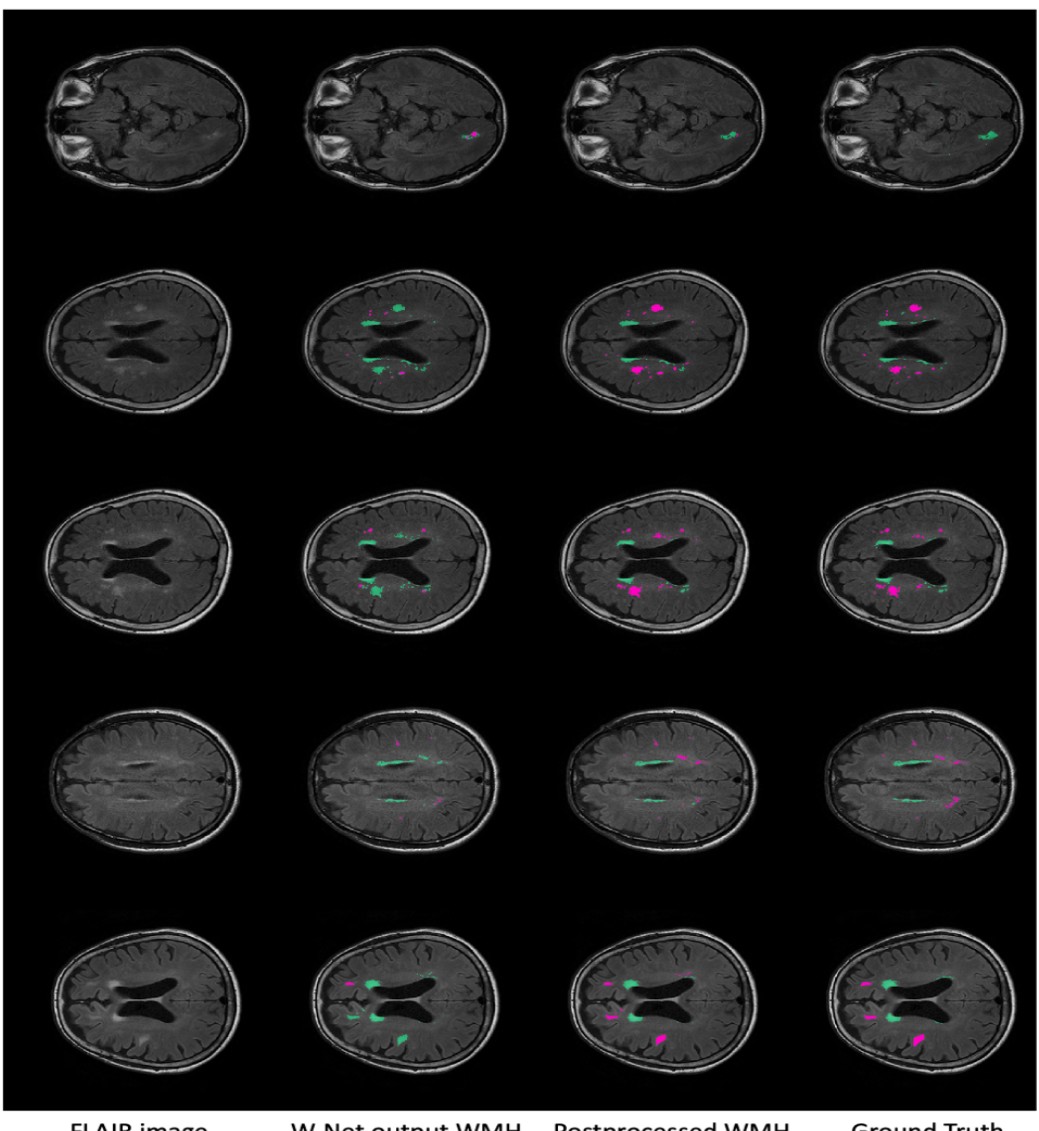

Figure 5: Results of the proposed method (green for PVWMH, pink for DWMH). W-NET output WMH : separately segmented output by two W-Net models, Postprocessed WMH : WMH mask which applied a fine-grained subdivision step.

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
