# OpenReview forum: "An Accurate and Efficient Segmentation Method of White Matter Hyper-intensity using Deep Learning"
_MIDL.io/2021/Conference — Submitted to MIDL 2021_

### Official Review · AnonReviewer4 · 2021-03-05

**Confidence:** 5
**Preliminary Rating:** 2
**Final Rating:** 2

**Summary:**

Authors present their work on WMH segmentation. Instead of segmenting all WMH at once, they train two separate W-nets to segment deep and periventricular WMH; which are then combined into the end result. This is a reasonable approach, because deep and periventricular WMH have different characteristics.

There are some additional results, including: lateral ventricle segmentation using a W-net and making the final division into deep/periventricular WMH via a connected-component based post-processing.

**Strengths:**

Considering deep and periventricular WMH separately is a nice approach. Authors compare this with segmenting all WMH at once and demonstrate that their approach indeed performs better.

This is a nice results. It is known in the WMH field (e.g. the WMH Segmentation Challenge by Kuijf etal) that segmenting the (often smaller and more rare) deep WMH is problematic. Treating this as a separate problem allows to develop more dedicated methods.

**Weaknesses:**

Some non-deep-learning methods on separating deep/periventricular WMH is missing, e.g. the work of Anbeek and Vincken. This could be added by the authors.

In the end, the distinction between deep/periventricular is not made using a neural network, but by a relative simple rule-based processing. In the Discussion, the authors mention the different rules that are used in clinic and which one they chose to implement (ventricle connectivity).

It seems that the deep/periventricular post-processing is not fully automatic, but the user can select a degree of expansion.

The deep/periventricular classification is not compared by the authors to the ground truth annotations they have made. Including this information would be valuable.

**Deanonymize Review:**

no

**Detailed Comments:**

The various rules that exist for classifying deep versus periventricular WMH could be included in the introduction. That also makes clear why authors chose this approach and not e.g. the 10mm rule.

It took me some time to understand that the authors are first making two W-nets to segment D/PV WMH, then combine it into a single WMH map, and finally use a rule-based system to make the final D/PV classification. That could be made more clear in the Introduction or Methods section.

Authors could also address why they take this approach and not use the results of the W-nets directly?

There are some repeating texts that can be removed to shorten the paper (e.g. about imbalanced data).

Do authors have information on intra/inter observer agreement on D/PV WMH segmentation?

Please rotate all figures so that Anterior is on top.

Authors should consider submitting their method to the WMH Segmentation Challenge. They include a comparison with the winning method, but the comparison is on different test sets. The challenge is still open and their approach (separately segmenting deep and periventricular WMH) could be very promising.

**Final Rating Justification:**

Authors have not addressed any reviewer comments before the rebuttal deadline. This is really unfortunate, because I have spent considerable time and effort to provide authors with a constructive review. Hence, I have to keep my preliminary rating as the final rating.

**Justification Of The Preliminary Rating:**

Authors present a very nice approach, but overall the methodological contribution is minimal. The essence of their approach is to split the data into deep/periventricular WMH and train two separate systems, instead of just one. Eventually, the final classification is not done using deep learning, but with a manual rule-based system.

**Paper Type:**

methodological development

**Questions To Address In The Rebuttal:**

Please include a comparison to the ground truth on the final deep / periventricular segmentation performance.
Please submit the method to the WMH Segmentation Challenge.
Please rewrite some parts of the manuscript to make the text more consise and clear for the reader.
Please rotate the figures.

**Special Issue:**

no

---

### Official Review · AnonReviewer2 · 2021-03-07

**Confidence:** 4
**Preliminary Rating:** 1

**Summary:**

This work aims to separate white matter hyperintensity into Deep WMHs and Periventricular WMHs. The authors aim to align with clinical criteria through the segmentation of the lateral ventricles. Two W-Net architectures are used for supervised segmentation. Evaluation has been done on 60, manually segmented FLAIR images.

**Strengths:**

the paper tries to align with clinical practice.
A pipeline is proposed.
Results may be en par with a MICCAI 2017 challenge winner.
--------------------------------------------------------------------

**Weaknesses:**

This paper does not seem to be ready yet. The authors claim that the main purpose of their work is to align with the clinical fazekas scale but they don't and leave it for future work.
It is unclear why it is difficult to compare to the winner of the MICCAI 2017 WMH segmentation challenge. Why has this method only been evaluated re recall and DICE?
It is unclear what actually has been achieved beyond yet another WMH segmentation algorithm. The initial aims of separating into Deep WMHs and Periventricular WMHs doesn't seem to have been addressed fully. The paper seems to overall propose a 1:1 implementation of some arbitrary clinical criteria in a processing pipeline aided by well known segmentation methods.

**Deanonymize Review:**

no

**Detailed Comments:**

the paper is difficult to read because of occasionally poor grammar and strange semantics. Nothing seriously wrong but cumbersome language could be condensed a lot to aid readability. "Because test dataset is different...", "we think that W-Net extracts image features more effectively than U-Net series...", "WMH segmentation is known as a class imbalanced problem." and many more.

**Justification Of The Preliminary Rating:**

This paper is not ready yet. It will need a considerable amount of work to make it suitable for a presentation at a conference like MIDL. Better evaluation and downstream-focused findings might make this work relevant for a clinical venue.

**Paper Type:**

validation/application paper

**Questions To Address In The Rebuttal:**

This paper needs more work to address the actual aim and claims need to be solidified. It needs to be shown that this method really reduces false negatives, that it is suitable for clinical scoring conversion and that results are valid, e.g. through cross-validation and this relatively small dataset.

**Special Issue:**

no

---

### Official Review · AnonReviewer1 · 2021-03-07

**Confidence:** 5
**Preliminary Rating:** 1
**Final Rating:** 2

**Summary:**

The paper first introduces a clinically relevant problem in WMH segmentation, i.e., dividing WMH into Deep WMHs (DWMH) and Periventricular WMHs (PVWMH).  The authors proposed to use a W-NET to perform segmentation with a post-processing step. The proposed method achieves a Dice coefficient score of 0.83 while separating WMH into PVWMH and DWMH.

**Strengths:**

1. Clinically relevant problem, i.e., dividing WMH into Deep WMHs (DWMH) and Periventricular WMHs (PVWMH).
2. Explore several strategies to handle the problem.
2. A simple yet effective post-processing step to separate Deep WMHs (DWMH) and Periventricular WMHs (PVWMH).

**Weaknesses:**

There are several weaknesses found when I went through the paper.

1. Evaluation metrics are not clear in terms of the relevant pathologies 'DWMH' and 'PVWMH'.

2. Some training details are missing. For example, is the segmentation for 'DWMH' and 'PVWMH' separated or not. The weight for the loss function is not mentioned.

3. Results need to be polished. Metrics are confusing when putting two-class (whole) results and three-class results together.


**Deanonymize Review:**

no

**Detailed Comments:**

There are several concerns when I went through the paper.

1. During the evaluation, are both DWMH and PVWMH considered as two classes when calculating the Dice and other metrics? How is the evaluation metric AUPRC calculated when there are two classes. I am asking this because as stated in the beginning by the authors, segmentation and separating them would be clinically relevant. But what are the criteria for a good separation? Are the precision and recall based on lesion-level or pixel-level?

2. The author mentioned that 'in the first step, convolutional neural networks (CNNs) are trained to separate PVWMH from DWMH segmentation, and both masks were combined into one whole WMH mask in the first stage.' Does it mean the segmentation of PVWMH and DWMH are done separately? If yes, why not training a network to segment them together and using the post-processing step proposed?

3. How to train the W-Net? Is it a coarse-to-fine architecture? Maybe a bit more details are needed. The W-Net contains two U-Nets and why is it more computationally efficient than a traditional U-Net?

4. And what's the weight for the weighted cross-entropy? Have the authors tried a Dice loss?

5. In the results section, a bit more explanation for the whole W-Net model is needed. Does it mean binary-class segmentation or three-class segmentation? I see that the authors mention 'whole WMH segmentation model (no separate)' in Table 1. However, how does this can be compared with the result of W-Net(PV/D) since one is binary-class, one is three-class (PV, D, and background). More importantly, this work focus on the separation PV/D, I didn't see any metric that evaluates the results considering the target pathologies. One more question, what's the ground truth when comparing the predictions in Table 1?
The result of training a three-class (PV/D/background) U-Net is expected.

6. Did the authors try U-Net + the proposed post-processing step?

Minor:
The title format for each section should be consistent. For example, Section 3.1 DATASET.

**Final Rating Justification:**

unfortunately, the authors did not submit a rebuttal to discuss.

**Justification Of The Preliminary Rating:**


1. Evaluation metrics are not clear in terms of the relevant pathologies 'DWMH' and 'PVWMH'.

2. Some training details are missing. For example, is the segmentation for 'DWMH' and 'PVWMH' separated or not. The weight for the loss function is not mentioned.

3. Results need to be polished. Metrics are confusing when putting two-class (whole) results and three-class results together.

**Paper Type:**

validation/application paper

**Questions To Address In The Rebuttal:**

Mostly from my comments above.

1. During the evaluation, are both DWMH and PVWMH considered as two classes when calculating the Dice and other metrics? How is the evaluation metric AUPRC calculated when there are two classes. I am asking this because as stated in the beginning by the authors, segmentation and separating them would be clinically relevant. But what are the criteria for a good separation? Are the precision and recall based on lesion-level or pixel-level?

2. The author mentioned that 'in the first step, convolutional neural networks (CNNs) are trained to separate PVWMH from DWMH segmentation, and both masks were combined into one whole WMH mask in the first stage.' Does it mean the segmentation of PVWMH and DWMH are done separately? If yes, why not training a network to segment them together and using the post-processing step proposed?

3. How to train the W-Net? Is it a coarse-to-fine architecture? Maybe a bit more details are needed. The W-Net contains two U-Nets and why is it more computationally efficient than a traditional U-Net?

4. And what's the weight for the weighted cross-entropy? Have the authors tried a Dice loss?

5. In the results section, a bit more explanation for the whole W-Net (and U-Net) model is needed. Does it mean binary-class segmentation or three-class segmentation? I see that the authors mention 'whole WMH segmentation model (no separate)' in Table 1. However, how does this can be compared with the result of W-Net(PV/D) since one is binary-class, one is three-class (PV, D, and background)? More importantly, this work focus on the separation of PV&D, I didn't see any metric that evaluates the results considering the two target pathologies. One more question, what's the ground truth when comparing the predictions in Table 1?

6. Did the authors try U-Net (whole) + the proposed post-processing step?

Minor:
The title format for each section should be consistent. For example, Section 3.1 DATASET.

**Special Issue:**

no

---

### Official Review · AnonReviewer3 · 2021-03-09

**Confidence:** 4
**Preliminary Rating:** 1

**Summary:**

The paper describes the segmentation of periventricular and deep white matter hyperintensity (PV/D-WMH) from T2 FLAIR images. Both types of WMH are separately extracted using two 2D W-net CNN. Their results are then merged and, a post-processing step is proposed to separate connected components in both classes based on their distance to the brain ventricles. 60 MRI scans are used as dataset.

**Strengths:**

The medical argument is convincing. The problem is interesting since it requires to account for intensity and geometry at the same time. A relevant geometrical criterion is proposed despite fuzzy or qualitative medical definition to discriminate both types of white matter.

**Weaknesses:**

Even though CNNs are used, the innovation is mostly in the post-processing stage. Furthermore, this stage is not evaluated: Two lines are missing in Table 1 with the performance to segment DWMH and PVWMH after the separation based on connected component analysis. The process that separately segments PV/D-WMH and then merges the results is intriguing and the author should assess the significance of the difference between the various models in Table 1.

**Deanonymize Review:**

no

**Detailed Comments:**

The proposed algorithm uses 2 W-net CNNs to separately extract the PVWMH and the DWMH from T2 FLAIR images. They are trained on a database of 60 patients (50/10 for train/validation). Some hyperparameters are used and the authors should give details about how they were set (e.g. alpha in equation 1). There does not appear to be any validation set to do that. Please comment.

The ventricles are segmented using a W-net trained to extract ventricles from T2 FLAIR images. It is trained with respect to ground truth automatic segmentation of T1 images registered to the T2 FLAIR images. It is not clear if the same database was used.

Then the ventricles masks are dilated. Please specify the width of this dilation. The text reads on p. 6 "The degree of expansion can be adjusted according to the subjectivity" which is unclear.

Then the two maps are merged and connected components are extracted. A connected component is labelled as PVWMH if it intersects the dilated ventricle mask, else if is labelled as DWMH. The merge step is very surprising, but the authors compare the result with a single WMH segmentation. The results appear slightly better but, given the close performance results and the small size of the database, significance scores should be computed to assess the validity of this comparison (see Table 1).

The proposed geometrical criterion makes sense, but it does not seem that surprising if a 2D segmentation approach is sub-optimal, since the 3D distance to the ventricle is key to separate both types of WMH. The authors seem to have an easy access to a variety of CNN architectures. Why was a 3D CNN not investigated? It also seems natural to use the 3D distance map to the ventricles as input (even as 2D slices) to the W-net CNNs. Why was not that considered? A visual attention mechanism also seems adapted here (or that is what the authors are looking for in a way). Why was not that experimented? All these comments and thoughts explain why I consider this paper to lack innovation and somewhat falls below state of the art research in deep learning.

**Justification Of The Preliminary Rating:**

Lack of innovation in the domain of deep learning. The choice for a post-processing step against more advanced deep learning approches needs a better arguent, including quantitative results for the proposed algorithm in terms of segmentation performance.


**Paper Type:**

validation/application paper

**Questions To Address In The Rebuttal:**

- provide details on the algorithms (hyperparameters, extent of dilation...)
- provide the performance in PV vs D-WMH detection after the proposed post-processing step, compared to the result obtained after the double CNN segmentation.
- a 3D CNN approach seems more natural since the 3D distance to the ventricles is critical for the discrimination. Why was that not investigated?
- alternatively, a natural way to incorporate distance information is to either use the 3D distance map to the ventricles as input (even with 2D CNNs) or to integrate it in other ways (e.g. visual attention). Why was that not investigated?

**Special Issue:**

no

---

### Meta-Review · Area_Chairs · 2021-03-31

**Recommendation:** Reject

**Metareview:**

all reviewers agree that the weaknesses of this work outweigh the few positive aspects and thus the paper cannot accepted in the current form. No rebuttal or revision was provided.

**Paper Type:**

validation/application paper

---

### Decision · Program_Chairs · 2021-03-31

Reject